# Genotype Distribution and Characteristics of Chronic Hepatitis C Infection in Estonia, Latvia, Lithuania, and Ukraine: The RESPOND-C Study

**DOI:** 10.3390/medicina59091577

**Published:** 2023-08-30

**Authors:** Ligita Jančorienė, Baiba Rozentāle, Ieva Tolmane, Agita Jēruma, Riina Salupere, Arida Buivydienė, Jonas Valantinas, Limas Kupčinskas, Jolanta Šumskienė, Eglė Čiupkevičienė, Arvydas Ambrozaitis, Olga Golubovska, Larysa Moroz, Robert Flisiak, Borys Bondar

**Affiliations:** 1Clinic of Infectious Diseases and Dermatovenerology, Institute of Clinical Medicine, Medical Faculty, Vilnius University, Vilnius University Hospital Santaros Klinikos, LT-08406 Vilnius, Lithuania; ligita.jancoriene@santa.lt (L.J.); arvydas.ambrozaitis@elnet.lt (A.A.); 2Latvian Centre of Infectious Diseases, Riga East Clinical University Hospital, LV-1006 Riga, Latvia; baiba.rozentale@aslimnica.lv (B.R.); ieva.tolmane@aslimnica.lv (I.T.); agita.jeruma@aslimnica.lv (A.J.); 3Faculty of Medicine, University of Latvia, LV-1586 Riga, Latvia; 4Faculty of Medicine, Riga Stradins University, LV-1007 Riga, Latvia; 5Tartu University Hospital, University of Tartu, EE-50406 Tartu, Estonia; riina.salupere@kliinikum.ee; 6Clinic of Gastroenterology, Nephrourology and Surgery, Vilnius University Hospital Santaros Klinikos, LT-08406 Vilnius, Lithuania; arida.buivydiene@santa.lt (A.B.); jonas.valantinas@santa.lt (J.V.); 7Centre of Hepatology, Gastroenterology and Dietetics, Vilnius University Hospital Santaros Klinikos, LT-08406 Vilnius, Lithuania; 8Department of Gastroenterology, Medical Academy, Lithuanian University of Health Sciences, LT-50161 Kaunas, Lithuania; j.sumskiene@gmail.com (J.Š.); egle.ciupkeviciene@lsmu.lt (E.Č.); 9Health Research Institute, Faculty of Public Health, Lithuanian University of Health Sciences, LT-47181 Kaunas, Lithuania; 10Infectious Disease Department, O.O. Bogomolets National Medical University, 01601 Kyiv, Ukraine; ogolubovska@gmail.com; 11Department of Infectious Diseases with the Course of Epidemiology, National Pirogov Memorial Medical University, 21018 Vinnytsya, Ukraine; larisa652002@yahoo.com; 12Department of Infectious Diseases and Hepatology, Medical University of Bialystok, 15-089 Bialystok, Poland; robert.flisiak1@gmail.com; 13AbbVie Biopharmaceuticals GmbH, 01032 Kyiv, Ukraine; borys.bondar@abbvie.com

**Keywords:** chronic hepatitis C, routes of infection, fibrosis stage, genotype, Baltic states, Ukraine

## Abstract

*Background and objectives:* Since 2013, highly effective direct-acting antiviral (DAA) treatment for chronic hepatitis C (CHC) has become available, with cure rates exceeding 95%. For the choice of optimal CHC treatment, an assessment of the hepatitis C virus (HCV) genotype (GT) and liver fibrosis stage is necessary. Information about the distribution of these parameters among CHC patients in Estonia, Latvia, and Lithuania (the Baltic states) and especially in Ukraine is scarce. This study was performed to obtain epidemiologic data regarding CHC GT and fibrosis stage distribution for better planning of resources and prioritization of patients for DAA drug treatment according to disease severity in high-income (the Baltic states) and lower-middle-income (Ukraine) countries. *Materials and methods:* The retrospective RESPOND-C study included 1451 CHC patients. Demographic and disease information was collected from medical charts for each patient. *Results:* The most common suspected mode of viral transmission was blood transfusions (17.8%), followed by intravenous substance use (15.7%); however, in 50.9% of patients, the exact mode of transmission was not clarified. In Ukraine (18.4%) and Estonia (26%), transmission by intravenous substance use was higher than in Lithuania (5%) and Latvia (5.3%). Distribution of HCV GT among patients with CHC was as follows: GT1—66.4%; GT3—28.1; and GT2—4.1%. The prevalence of GT1 was the highest in Latvia (84%) and the lowest in Ukraine (63%, *p* < 0.001). Liver fibrosis stages were distributed as follows: F0—12.2%, F1—26.3%, F2—23.5%, F3—17.1%, and F4—20.9%. Cirrhosis (F4) was more prevalent in Lithuanian patients (30.1%) than in Estonians (8.1%, *p* < 0.001). *Conclusions:* This study contributes to the knowledge of epidemiologic characteristics of HCV infection in the Baltic states and Ukraine. The data regarding the patterns of HCV GT and fibrosis stage distribution will be helpful for the development of national strategies to control HCV infection in the era of DAA therapy.

## 1. Introduction

An estimated 58 million people have chronic hepatitis C virus infection, with about 1.5 million new infections occurring per year. The highest burden of disease is in the eastern Mediterranean region and European region, with 12 million people chronically infected in each region [1,2]. In the Southeast Asia region and the western Pacific region, an estimated 10 million people in each region are chronically infected. Nine million people are chronically infected in the African region and 5 million in the region of the Americas. In the European region of the World Health Organization (WHO), the prevalence of HCV is higher outside the EU/EFTA countries, and many infected individuals are not aware of their infection [3]. HCV is most commonly transmitted via the blood. Thus, intravenous (as well as intranasal) substance users, dialysis patients, and those undergoing tattooing or manicure and pedicure services are at risk of infection [4,5]. In rare cases, HCV can also be transmitted vertically from a mother to a child, and the virus can also be spread through sexual intercourse [4,5]. Of those with chronic HCV infection, the risk of cirrhosis ranges from 15% to 30% within 20 years [6,7]. Six major genotypes of HCV have been defined, with more than 50 subtypes described [8]. The most common subtypes found in European countries are 1a and 1b in genotype 1 and 2a and 2b in genotype 2. These variants have become very widely distributed over the past decades as a result of transmission through blood transfusion and various other invasive medical procedures and by needle sharing between intravenous substance users. They now represent the vast majority of infections encountered clinically.

Determination of HCV genotype and liver fibrosis stage is essential to making treatment decisions, as the regimens, dosing, and duration of therapy vary across the genotypes. [9]. Access to HCV treatment is improving but remains too limited. Of the 58 million persons living with HCV infection globally in 2019, an estimated 21% (15.2 million) knew their diagnosis, and of those diagnosed with chronic HCV infection, around 62% (9.4 million) persons had been treated with DAAs by the end of 2019.

The prevalence of HCV infection is relatively high in the Baltic countries. In Estonia, it affects 1.5% to 2.0% of the population [10], in Latvia, 2.4% [11], and in Lithuania, 1.7% [12]. In Ukraine, official statistics or registries for HCV infection are scarce; however, the estimated prevalence of infection is assessed as very high—3.5% [13].

In 2016, the World Health Assembly passed a resolution to eliminate viral hepatitis as a public health threat by 2030 [14]. Safe and effective drugs, direct-acting antivirals (DAA), are now available; however, most patients remain unaware of their infection, which may be recognized only in the late stages when the complications have occurred. Furthermore, the high cost of new drugs requires prioritization in their allocation. Thus, identifying patients at high risk of dying from CHC and ensuring appropriate treatment is a healthcare challenge, especially in lower-income countries. WHO issued new guidelines for treating CHC based on a patient’s clinical history, as well as on the genotype of HCV [15]. WHO now recommends that testing, care, and treatment for persons with chronic hepatitis C infection can be provided by trained nonspecialist doctors and nurses.

Determination of HCV genotype and liver fibrosis stage is essential to making treatment decisions. Individuals with advanced fibrosis are more likely to develop complications that require urgent treatment. In addition, they require disease management interventions such as hepatocellular carcinoma (HCC) screening. Clinical data regarding disease progression and fibrosis stage distribution are limited in the Baltic states and Ukraine. Pan-genotypic DAAs remain expensive in many high- and upper-middle-income countries. Proper resource planning is especially important in Ukraine, where the war has created many challenges for the healthcare system.

This study aimed to evaluate the distribution of genotypes and stages of fibrosis among CHC patients for better planning of healthcare resources to achieve the WHO goal to eliminate hepatitis C infection by 2030 in the high-income Baltic countries and the lower-middle-income country of Ukraine.

## 2. Materials and Methods

### 2.1. Study Design and Sample

This was an epidemiologic study with a retrospective and multi-center design. Centers from different geographical regions of Ukraine, Estonia, Latvia, and Lithuania were selected to participate in the study based on the following selection criteria: selected centers were both inpatient and outpatient reference centers with experience in the treatment of CHC and current access to patients with HCV infection. Centers also had to have the ability to conduct the study following applicable legal and regulatory requirements, as well as to perform data entry via an online electronic case report form (e-CRF). Centers had to represent the estimated number of patients in rural or urban areas. Altogether, 15 centers (8 in Ukraine, 1 in Latvia, 3 in Lithuania, and 3 in Estonia) were involved in the study. The study was conducted from 28 July 2015 to 20 April 2016.

Study enrollment was planned for approximately 1450 patients (100 from Estonia, 150 from Latvia, 200 from Lithuania, and 1000 from Ukraine). Study investigators enrolled patients until the sample had reached these predefined limits. To reduce selection bias at the patient level, all patients with HCV infection who attended a routine visit at each participating center within the past 6 months before study initiation were identified as possible eligible patients for the study. Furthermore, patients had to fulfil the following selection criteria: diagnosis of chronic HCV infection confirmed with a positive HCV-RNA test, defined HCV GT, and age ≥18 years. All patients had to provide patient authorization (consent) for the use/disclosure of data. Patients with a diagnosis of acute hepatitis C were excluded from data collection.

In total, 1490 patients were enrolled in the study. Due to missing data, 39 (2.6%) of patients were excluded from the study. Therefore, data from 1451 patients were analyzed (100 from Estonia, 150 from Latvia, 201 from Lithuania, and 1000 from Ukraine).

### 2.2. Data Collection and Variables

Information was collected from the patient’s medical documentation on a pre-established e-CRF at a single time point for each patient. Demographic data included age at the time of the study, gender, geographic location (country), duration of HCV infection, GT classification, alcohol consumption, and substance use. Patients were asked about their alcohol consumption habits, which were classified as ‘Yes’ if they consumed any alcoholic drink or ‘No’ if they did not. Regarding substance use, patients were classified as former drug users, active drug users, and those who had never used drugs. Patients who failed to provide an answer about their substance or alcohol consumption were marked as ‘Unknown’. Collected clinical data were fibrosis stage, HCC status, and infection characteristics.

GTs were grouped as follows: GT1 (overall and separately as GT1a, GT1b, and GT1 undefined), GT2, GT3, GT4, GT5, GT6, GT other, or GT mixed.

The liver fibrosis stage was most frequently based on histology results obtained from liver biopsy specimen analysis and/or liver stiffness measurement by transient elastography (TE) using a FibroScan^®^ model 402 with an M probe (Echosens, Paris, France). According to the manufacturer’s recommendations, only TE results obtained with 10 valid measurements and a success rate of at least 60% (with <30% interquartile range) were considered reliable. Fibrosis stages were categorized using the METAVIR scoring system as follows: F0—no fibrosis; F1—portal fibrosis without septa; F2—portal fibrosis with few septa; F3—numerous septa without cirrhosis; and F4—cirrhosis.

In this study, the following self-reported modes of infections were assessed: unsafe injection (e.g., substance use and needle stick injury), blood transfusion, hemodialysis, organ transplantation, dental procedures, other invasive procedures (e.g., tattooing/body piercing), unknown, and other.

As HCV infection shares a similar mode of transmission with other viruses, patients were also categorized according to their co-infection status with human immunodeficiency virus (HIV) infection.

### 2.3. Data Analysis

The categorical variables were presented as proportions and compared using an χ^2^ test and Z-test with Bonferroni correction. The Bonferroni-corrected alpha level was set at 0.008 when proportions between 4 countries were compared using the Z-test (0.05/6 pairwise comparisons). Statistical analysis was performed using SAS 9.2 for Windows (SAS Institute, Cary, NC, USA).

### 2.4. Ethical Statement

All subjects gave their informed consent for inclusion before they participated in the study. The study was conducted in accordance with the Declaration of Helsinki, and approval to conduct the study was issued by the Ethics Committee of Latvia, protocol N 19-A/15 on 6 August 2015, the Lithuanian Bioethics Committee, protocol NL-15-07/1, and the Tallinn Medical Researches Ethics Committee, protocol N 1139, on 17 September 2015. In Ukraine, the legislation does not require ethical approval for noninterventional studies.

## 3. Results

### 3.1. Demographics and General Characteristics

The study population consisted of 845 (58.2%) men and 606 (41.8%) women. The mean age of patients was 43.9 (SD 11.7) years (range from 18.0 to 82.0 years). The general characteristics of the study population are shown in Table 1.

Substance abuse was uncommon among study participants. There were 20 (2%) active drug users in Ukraine and 1 (0.5%) in Lithuania. The highest proportion of former drug users was found in Estonia (24%) and Ukraine (17.1%). The prevalence of alcohol consumption varied from 3.4% in Latvia to 19% in Estonia. The suspected mode of infection differed between countries. In Lithuania, the main route of HCV transmission was blood transfusion (26.4%). Intravenous substance use was most frequently recorded in Estonia (26%) and Ukraine (18.4%). Even 34% of Latvian patients were suspected of being infected via dental procedures. The route of transmission was not defined in 738 (50.9%) patients.

The highest proportion of patients with a disease duration of 10 years and more was in Estonia (33%). In all countries except Estonia, the duration of disease in most patients was up to 3 years. HIV and/or HBV co-infection was identified in 125 (12.5%) patients in Ukraine and 24 (24%) in Estonia.

Overall, comorbidities were reported in 30.7% of patients (445/1451). Other liver diseases accounted for 55.5% of all comorbidities (208/445 patients), and they were the most frequent conditions in every country. Other chronic infectious diseases made up 16.3% of comorbidities (61/445) and 11.5% of diseases of the circulatory system (43/445) (data are not shown).

In all countries except Lithuania, the highest proportion of patients had an F1 stage of liver fibrosis. Almost one-third (30.1%) of Lithuanian patients were diagnosed with liver cirrhosis (F4). HCC was found in 4 (0.4%) patients in Ukraine and 8 (1.8%) patients in the Baltic countries.

### 3.2. Genotype Distribution

As shown in Table 2, the overall pattern of HCV GT distribution was similar among all studied countries. In general, GT1 was most common, accounting for 963 of all infections (66.4%), followed by GT3 (*n* = 408; 28.1%). Only small percentages of GT2, mixed GT, or GTs not further classified were reported (Table 2). GTs 4, 5, and 6 were not identified among the study participants. Subtype 1b accounted for 76.4% (765/963) of all GT1 infections and 94.3% (765/811) of all subtype-defined GT1 infections.

Despite the predominance of GT1 in all studied countries, some regional differences in GT distribution have been found (Table 2). The highest proportion of GT1 was observed in Latvia—84% (*p* < 0.001 vs. Ukraine; *p* = 0.001 vs. Lithuania and *p* = 0.002 vs. Estonia). At the same time, the proportion of GT3 in Latvia was the lowest—14% (*p* < 0.001 vs. Ukraine, *p* = 0.040 vs. Lithuania and *p* = 0.034 vs. Estonia).

Some variations in GT predominance were identified, depending on disease characteristics. GT1 was detected more frequently in those >50 years of age (74.5% vs. 62.3%, *p* < 0.01), but GT3 was found more often for those aged < 50 years (32.9% vs. 18.0%, *p* < 0.01).

GT distribution differed by fibrosis stages (Table 3). The prevalence of GTs was compared in two groups of patients—those with mild liver disease (F0–F1) and those with more advanced disease (F2–F4). The proportion of GT1 among F2–F4 patients (71.8%; 510/710) was significantly higher than in the F0–F1 group (63.1%; 281/445; *p* = 0.002). The proportion of GT2 among F2–F4 patients was significantly lower than in the F0–F1 group (2.4% (17/710) vs. 6.3% (28/445), respectively; *p* = 0.002). GT3 was found equally often in both groups (24.4% and 29.2%, respectively).

GT1 was determined more often among patients with than without HIV co-infection (47.7% and 68.5%, respectively; *p* < 0.001) (Table 3). The prevalence of GT1 and GT3 differed between intravenous substance users and those who assumed blood transfusion was the mode of infection. GT1 was found in 44.3% of intravenous substance users and in 81.4% of patients suspected of being infected through blood transfusion (*p* < 0.001). Meanwhile, GT3 prevalence in intravenous substance users was much higher than in those probably infected through blood transfusion (50.9% and 14.0%, respectively, *p* < 0.001).

## 4. Discussion

A better knowledge of HCV infection epidemiology in a particular country, including the distribution of the various GTs, may substantially contribute to the effective prevention and treatment of CHC by focusing on people at risk of infection. This study was one of the first to describe the current epidemiologic features of HCV infection in the Baltic countries and Ukraine.

Our study demonstrated that the distribution of the suspected modes of infection varied between countries. The traditional routes of HCV transmission have changed over the last decades. The introduction of screening assays in 1990 reduced the risk of transmitting HCV via blood transfusions [16]. In our study, blood transfusion was reported as the suspected road of infection by 17.8% of patients, more often in Lithuania. Iatrogenic transmission of HCV because of surgical interventions or dental treatment also declined globally [17]. According to our data, only in Latvia were dental procedures recorded as the suspected mode of infection for every third patient. Strict adherence to standard safety procedures is required to prevent iatrogenic transmission of HCV [18].

Intravenous substance use is an important risk factor for transmission of HCV [19,20]. The majority of new infections are related to illicit substance use [20]. This is of particularly high concern, as a large number of active drug users are younger than 25 years. More people who inject drugs had HCV than HIV infection [19]. Our study found the highest proportion of former drug users in Estonia and Ukraine. Effective preventive strategies, substance abuse treatment, and access to sterile injection equipment may help to reduce the prevalence of HCV infection among substance users; however, the evidence suggests the suboptimal level of implementation of preventive programs in Eastern European countries [21].

HCV infection contributes to liver cirrhosis and HCC [6,22]. Due to the asymptomatic nature of HCV infection, the majority of patients are not aware of when they have been infected [23]. HCV infection is frequently diagnosed when it becomes chronic and end-stage liver diseases occur. Our data revealed that every third patient in Lithuania and every fifth in Ukraine and Latvia were diagnosed with liver cirrhosis. HCC was found in 12 (0.8%) patients, more in the Baltic countries than in Ukraine. However, even 14.3% of patients were not screened for cancers.

Our study found that the distribution of HCV genotypes in Estonia, Latvia, Lithuania, and Ukraine is similar to in Western countries [8]. Several studies analyzed the role of HCV genotypes in the prediction of liver disease progression and found that HCV genotype 1b increased the risk of hepatocellular carcinoma [24,25]. Our data demonstrated the high prevalence of GT1 HCV (66.4%) in the Baltic countries and Ukraine. Subtype 1b dominated among all GT1 infections. The distribution of HCV genotypes in our study was consistent with previously reported data from Estonia, Latvia, Lithuania, and Ukraine [26]. The proportion of GT1 was higher in patients from Latvia and older patients in all countries. Previous studies also found that genotype distribution is related to age [27,28]. GT1 was more frequently observed in older age groups. Our data revealed a higher prevalence of GT1 in patients with more advanced liver fibrosis stages. These findings appear to be consistent with previous observations that showed an association between the HCV GT and the risk of end-stage liver diseases [24,29]. On the contrary, some other studies demonstrated that GT3 is associated with a higher rate of hepatic steatosis and more rapid progression of liver fibrosis compared to infection with other HCV GTs [30,31]. An increased risk of HCC among GT3-infected individuals than among those with other GTs was reported in the USA and Korea [32,33]. We were not able to analyze the association between HCC and GTs because of the very low number of patients with HCC.

The outcome of a patient depends on an appropriate treatment regimen, which is determined by various factors, including GT. The introduction of DAA has increased the efficacy of HCV infection treatment; however, the GT3 infection showed a lower response compared to other GTs, particularly in treatment-experienced patients and those with liver cirrhosis [34]. The European Association for the Study of the Liver recommends assessment of liver disease severity and HCV genotype determination before therapy. Pan-genotypic HCV drug regimens can be used to treat individuals without identifying their HCV genotype and subtype; however, identifying certain genotypes may be required where drug pricing dictates genotype-specific treatment and, also, to optimize treatment regimens [35].

The availability of DAA and treatment with their combinations can cure the majority of HCV-infected patients; however, a high proportion of them remain still undiagnosed and are at risk of developing cirrhosis and HCC [1,2]. In addition, some limitations in access to treatment still exist in the countries, particularly in those with lower income [36]. The war has created many challenges for Ukraine’s healthcare system, including infection control. Due to the difficult conditions, especially in a war zone, the possibility of contracting HCV and other blood-borne infections increases. Therefore, the prevalence of those diseases may increase in the future. As a result of the war, the reduction in healthcare resources in Ukraine has created an urgent need to develop a system for the prioritization of patients for DAA treatment. A greater risk of HCV transmission and lack of healthcare resources may have a negative effect on the elimination efforts of HCV infection.

Our study provided new data on the HCV epidemiological situation in the Baltic countries and Ukraine; however, several limitations of the study also should be mentioned. The study design was cross-sectional; therefore, causal links cannot be established. The sample sizes were relatively small, especially in the Baltic countries. A larger sample was studied in Ukraine because of a particular lack of data on the epidemiological situation of HCV in the country. Finally, quite a large proportion of the medical records lacked information on some of the variables analyzed.

## 5. Conclusions

This study contributes to the knowledge of epidemiologic characteristics of HCV infection in the Baltic states and Ukraine. The data regarding the patterns of HCV GT and fibrosis stage distribution will be helpful for the development of national strategies to control HCV infection in the era of DAA therapy. The national programs for HCV elimination, which include unlimited access to effective treatment regimes, nationwide screening programs to diagnose hidden cases of HCV infection in the population, and appropriate preventive measures to reduce the incidence of HCV infection, are urgently needed to achieve the WHO target for HCV elimination by 2030.

## Figures and Tables

**Table 1 medicina-59-01577-t001:** General characteristics of the study participants.

Characteristics	Ukraine (N = 1000)		Estonia (N = 100)		Latvia (N = 150)		Lithuania (N = 201)		Total (N = 1451)		*p*-Value
	n	%	n	%	n	%	n	%	n	%	
**Male**	613	61.3	59	59.0	69	46.0	104	51.7	845	58.2	<0.001
**Female**	387	38.7	41	41.0	81	54.0	97	48.3	606	41.8	
**Drug abuse**											<0.001
Former drug user	171	17.1	24	24.0	11	7.3	10	5.0	216	14.9	
Active drug user	20	2.0	0	0.0	0	0.0	1	0.5	21	1.5	
Never used	772	77.2	70	70.0	137	91.4	188	93.5	1167	80.4	
Unknown	37	3.7	6	6.0	2	1.3	2	1.0	47	3.2	
**Alcohol consumption**											<0.001
Yes	175	17.5	19	19.0	5	3.4	30	14.9	229	15.8	
No	757	75.7	67	67.0	143	95.3	170	84.6	1137	78.3	
Unknown	68	6.8	14	14.0	2	1.3	1	0.5	85	5.9	
**Suspected mode of infection**											<0.001
Intravenous drug use	184	18.4	26	26.0	8	5.3	10	5.0	228	15.7	
Blood transfusion	156	15.6	21	21.0	28	18.7	53	26.4	258	17.8	
Dental procedures	172	17.2	2	2.0	51	34.0	2	1.0	227	15.6	
Unknown	488	48.8	51	51.0	63	42.0	136	67.6	738	50.9	
**Duration of HCV infection**											<0.001
≥0 and <3 years	410	41.0	22	22.0	50	33.4	78	38.8	560	38.7	
≥3 and <5 years	211	21.2	19	19.0	17	11.3	32	15.9	279	19.2	
≥5 and <10 years	270	27.0	26	26.0	44	29.3	52	25.9	392	27.0	
≥10 years	108	10.8	33	33.0	39	26.0	39	19.4	219	15.1	
**Fibrosis stage**											<0.001
F0	122	16.6	6	6.9	12	8.5	1	0.5	141	12.2	
F1	178	24.4	37	42.5	58	40.8	31	16.1	304	26.3	
F2	171	23.3	24	27.6	21	14.8	55	28.4	271	23.5	
F3	113	15.4	13	14.9	24	16.9	48	24.9	198	17.1	
F4 (cirrhosis)	149	20.3	7	8.1	27	19.0	58	30.1	241	20.9	
Total	733	100	87	100	142	100	193	100	1155	100	
**HIV and/or HBV co-infection status**											0.001
Yes	125	12.5	24	24.0	0	0.0	0	0.0	149	10.3	
No	875	87.5	76	76.0	150	100	201	100	1302	89.7	
**HCC status**											<0.001
Yes	4	0.4	1	1.0	3	2.0	4	2.0	12	0.8	
No	842	84.2	82	82.0	120	80.0	187	93.0	1231	84.9	
Unknown	154	15.4	17	17.0	27	18.0	10	5.0	208	14.3	

Abbreviations: HCC, hepatocellular carcinoma; HCV, hepatitis C virus.

**Table 2 medicina-59-01577-t002:** HCV genotype distribution according to geographic location.

Country	Genotypes								
	GT1 All Subtypes, *n* (%)	GT1 Subtype *			GT2, *n* (%)	GT3, *n* (%)	Other than GT1–6, *n* (%)	Mixed GTs, *n* (%)	*p*-Value
		GT1 Undefined, *n* (%)	GT1a, *n* (%)	GT1b, *n* (%)					
Ukraine	630	88	19	523	45	317	2	6	<0.001
*n* = 1000	(63.0%)	(14.0%)	(3.0%)	(83.0%)	(4.5%)	(31.7%)	(0.2%)	(0.6%)	
Latvia	126	51	1	74	2	21	0	1	
*n* = 150	(84.0%)	(40.5%)	(0.8%)	(58.7%)	(1.3%)	(14.0%)		(0.7%)	
Lithuania	140	13	23	104	10	45	0	6	
*n* = 201	(69.7%)	(9.3%)	(16.4%)	(74.3%)	(5.0%)	(22.3%)		(3%)	
Estonia	67	0	3	64	3	25	0	5	
*n* = 100	(67.0%)		(4.5%)	(95.5%)	(3.0%)	(25.0%)		(5.0%)	
Total	963	152	46	765	60	408	2	18	
*n* = 1451	(66.4%)	(15.8%)	(4.8%)	(79.4%)	(4.1%)	(28.1%)	(0.1%)	(1.3%)	

Abbreviations: GT, genotype; HCV, hepatitis C virus. * Percentages of subtypes within GT1 are shown.

**Table 3 medicina-59-01577-t003:** HCV genotype distribution according to patient characteristics.

Characteristic	Genotypes					
	GT1, *n* (%)	GT2, *n* (%)	GT3, *n* (%)	Other Than GT1–6, *n* (%)	Mixed GTs, *n* (%)	*p*-Value
**Fibrosis stage**						
F0	92 (65.2%)	6 (4.3%)	42 (29.8%)	0	1 (0.7%)	0.038
F1	189 (62.2%)	22 (7.2%)	88 (29.0%)	1 (0.3%)	4 (1.3%)	
F2	194 (71.6%)	9 (3.3%)	63 (23.3%)	0	5 (1.8%)	
F3	147 (74.3%)	2 (1.0%)	47 (23.7%)	0	2 (1.0%)	
F4 (cirrhosis)	169 (70.2%)	6 (2.5%)	63 (26.1%)	0	3 (1.2%)	
**HIV Co-infection status**						
Yes	71 (47.7%)	3 (2.0%)	70 (47.0%)	0	5 (3.3%)	<0.001
No	892 (68.5%)	57 (4.3%)	338 (26.0%)	2 (0.2%)	13 (1.0%)	
**Suspected mode of infection**						
Intravenous drug use	101 (44.3%)	5 (2.2%)	116 (50.9%)	0	6 (2.6%)	<0.001
Blood transfusion	210 (81.4%)	12 (4.6%)	36 (14.0%)	0	0	
Invasive healthcare mode *	172 (69.4%)	6 (2.4%)	69 (27.8%)	0	1 (0.4%)	
Other	206 (68.0%)	16 (5.3%)	77 (25.4%)	1 (0.3%)	3 (1.0%)	
Unknown	274 (66.2%)	21 (5.1%)	110 (26.6%)	1 (0.2%)	8 (1.9%)	

Abbreviations: GT, genotype; HCV, hepatitis C virus. * Invasive healthcare mode includes dental procedures, organ transplantation, and needle stick injuries.

## Data Availability

The data presented in this study are available on request from the corresponding author. The data are not publicly available because of ethical issues.

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
