# Peer review of "Genotype Distribution and Characteristics of Chronic Hepatitis C Infection in Estonia, Latvia, Lithuania, and Ukraine: The RESPOND-C Study"

_medicina, 2023, doi:10.3390/medicina59091577_

Round 1
Reviewer 1 Report
Thank you for inviting me to review the manuscript titled "Retrospective cross-sectional study on chronic hepatitis C in Estonia, Latvia, Lithuania, and Ukraine: virus, patient, and disease characteristics in patients under care (RESPOND-C study)".
This study provides valuable insights into the epidemiological characteristics of chronic HCV infection in Estonia, Latvia, Lithuania, and Ukraine. However, there are some scientific writing issues that should be addressed by the authors before publication. Here are my comments for the authors:
Title:
- The study type does not need to be mentioned in the title.
- If the study name is not mentioned in the methods, it should not be included in the title either.
- Suggested title: "Genotype distribution and characteristics of chronic hepatitis C infection in Estonia, Latvia, Lithuania, and Ukraine"
Abstract:
- Instead of using "Baltic states," it is better to explicitly name the countries.
- Select more appropriate keywords from the PubMed MeSH database.
Introduction:
- Provide existing evidence on genotype distribution in the area and highlight any differences compared to the rest of the world.
- Clearly indicate the importance and novelty of this study.
Methods:
- "Retrospective cross-sectional study" is not correct. A study cannot be both retrospective and cross-sectional.
- Specify the time period during which data collection took place.
- Mention the laboratory methods used for genotype testing.
- State that the "self-reported" method was used to assess the route of transmission.
- Specify the laboratory techniques used to evaluate HBV and/or HIV coinfection.
Results:
- Replace the stigmatizing phrases "drug abuse" and "drug user" with less stigmatizing alternatives in both the text and the table. Suggestions: "substance use," "former drug use," and "active drug use."
- Change "injection drug use" to "intravenous drug use."
Discussion:
- Discuss the genotype distribution in the study area and compare it with the rest of the world.
- Discuss the potential impact of the war in Ukraine on the epidemiology of HCV infection and other blood-borne infections in the future.
- Consider discussing the effect of the war on HCV elimination efforts in Ukraine, as it could be of interest to readers.
Overall:
- The manuscript requires language editing to improve readability and clarity.
Author Response
Dear Reviewer,
We would like to thank you for your efforts and time to read and revise our manuscript. We appreciate your comments and suggestions. We hope that we have successfully addressed all of the concerns raised, and we believe that the manuscript has been substantially improved. Our detailed responses to the comments and the description of the changes we have made to the manuscript are provided in the uploaded file.

Reviewer 2 Report
In this manuscript, the authors investigated the epidemiologic data regarding hepatitis C virus (HCV) genotype (GT) and fibrosis stage distribution of chronic hepatitis C (CHC) patients in Baltic states and Ukraine. 1451 CHC patients were recruited. The authors found that the most frequent suspected mode of viral transmission was blood transfusions, followed by injection drug use, that the exact mode of transmission was not clarified in 50.9% of patients, that distribution of HCV GT was as follows: GT1, 66.4%; GT3, 28.1; and GT2, 4.1%, and that the liver fibrosis stages were distributed as follows: F0, 12.2%, F1, 26.3%, F2, 23.5%, F3, 17.1% and F4, 20.9%. The authors concluded that this study demonstrated the patterns of HCV GT and fibrosis stage distribution of HCV infection in the Baltic states and Ukraine, and suggested that these results will be helpful for the development of national strategies to control HCV infection in the era of DAA therapy.
This is an epidemiologic, retrospective, cross-sectional study to investigate the suspected transmission route, HCV genotype and fibrosis stage distribution of chronic hepatitis C patients in Baltic states and Ukraine. The data were well collected and analyzed. The manuscript was well prepared. Although the originality was not high, this article can provide useful information for the readers to understand the epidemiologic situation of HCV infection in Baltic states and Ukraine.
Comments
1. The Title of reference 12 was missed.
2. The English needs polishing.
Needs polishing
Author Response

(The authors gave the same response as above.)

Reviewer 3 Report
Title:
Title is too long.
A good title is typically around 10 to 15 words
Introduction:
This section needs improvement;
· Unsafe therapeutic injection practices including reuse of syringes and needles is a common cause of Hepatitis C transmission in some countries.
It needs to be mentioned in introduction
· Acknowledge work of other researchers, done on the topic and identify the gaps knowledge which need to be filled
· Describe clearly General and Specific objectives of the study
Methods:
This is section also needs improvement;
When the study was conducted ? Start and end dates ?
· Selection of treatment centers
Why the treatment centers were not randomly selected ? Non random selection of treatment centers might have introduced selection bias in the study . Please justify
· How sample size of 1450 was calculated ? Describe the sampling frame from which sample was drawn
· Why non random sampling method was followed ?
· Describe the method how genotyping of HCV were done
· The study design was retrospective cross sectional study and data was conducted from clinical files of the patients, how informed consent was obtained from study participants
· How confidentiality issues were addressed?
Results:
Summarize the results according to objectives of the study
Discussion:
Discuss main findings of the study, comparing results of your study with similar published studies
· Describe strengths and weaknesses of your study
Conclusion:
Restate your research problem, describe your main findings and give interpretations of the results
no comments
Author Response

(The authors gave the same response as above.)

Round 2
Reviewer 1 Report
I appreciate the authors' work to revise their manuscript. All the peer-review comments are addressed in this revision.